# Ultraviolet supercontinuum generation driven by ionic coherence in a strong laser field

Hongbin Lei [1], Jinping Yao [2✉], Jing Zhao [1], Hongqiang Xie [2,3], Fangbo Zhang [2], He Zhang[2], Ning Zhang [2], Guihua Li [4], Qian Zhang [1], Xiaowei Wang[1], Yan Yang[1], Luqi Yuan [5], Ya Cheng [2] & Zengxiu Zhao [1✉]

Supercontinuum (SC) light sources hold versatile applications in many fields ranging from imaging microscopic structural dynamics to achieving frequency comb metrology. Although such broadband light sources are readily accessible in the visible and near infrared regime, the ultraviolet (UV) extension of SC spectrum is still challenging. Here, we demonstrate that the joint contribution of strong field ionization and quantum resonance leads to the unexpected UV continuum radiation spanning the 100 nm bandwidth in molecular nitrogen ions. Quantum coherences in a bunch of ionic levels are found to be created by dynamic Stark-assisted multiphoton resonances following tunneling ionization. We show that the dynamical evolution of the coherence-enhanced polarization wave gives rise to laser-assisted continuum emission inside the laser field and free-induction decay after the laser field, which jointly contribute to the SC generation together with fifth harmonics. As proof of principle, we also show the application of the SC radiation in the absorption spectroscopy. This work offers an alternative scheme for constructing exotic SC sources, and opens up the territory of ionic quantum optics in the strong-field regime.

[1] Department of Physics, National University of Defense Technology, Changsha, China. [2] State Key Laboratory of High Field Laser Physics, Shanghai Institute of Optics and Fine Mechanics, Chinese Academy of Sciences, Shanghai, China. [3] School of Science, East China University of Technology, Nanchang, China. [4] School of Science, East China Jiaotong University, Nanchang, China. [5] State Key Laboratory of Advanced Optical Communication Systems and Networks, School of Physics and Astronomy, Shanghai Jiao Tong University, Shanghai, China. ✉email: jinpingmrg@163.com; zhao.zengxiu@gmail.com

The control of light-matter interaction plays a fundamental role in many frontier disciplines, which not only helps modify the macroscopic properties of matter[1–3], but also enables the creation of state-of-art light sources[4–7]. Generally, there are two benchmark approaches to reinforce or manipulate light-matter interaction processes. The quantum optics approach usually relies on energy exchange between atoms and the driving laser field via resonant excitation, where photonic aspect and quantum nature are indispensable[8]. The second approach, i.e., the strong field approach, takes advantage of an intense femtosecond laser to hammer media nonperturbatively. Because as many as hundreds of photons could be simultaneously involved during the fleeting transition, the electron dynamics can be well understood within the semiclassical concepts of trajectory and scattering[9,10], giving birth to strong field physics. Particularly, within the widely applied strong-field approximation, the electronic ground and continuum states are well considered whereas excited states are ignored[11]. Over the past decades, interdisciplinary studies of strong field physics and quantum optics have been rarely explored owing to fundamental differences in their focused subjects and the used laser parameters.

Until recently, special attentions have been concentrated on atomic excited states prepared by a strong laser field[12–23]. These findings emphasize the crucial role of quantum coherence in strong field processes[12,15–18,20] and open up promising applications in spectral compression[12,13,22,23], space-time control of extreme ultraviolet radiation[18] and amplification of light fields[19]. However, these investigations are mainly carried out in neutral atoms. As compared to neutral atoms, the molecular ion prepared by ultraintense ultrashort laser pulses is a unique non-equilibrium quantum system, which not only possesses rich vibrational and rotational levels as well as suitable electronic energy separation, but also can survive at a higher laser intensity. These properties provide an unprecedented opportunity to investigate quantum optics in the strong-field framework.

In this work, we find and experimentally confirm the generation of the unexpected ultraviolet (UV) continuum radiation spanning the 100 nm bandwidth in strong-field-ionized nitrogen molecules. The supercontinuum (SC) generation is the result of joint contribution of strong field ionization and quantum coherence among various ionic states, which is intrinsically different from prior mechanisms for spectral broadening[24,25]. Our work bridges the gap of strong-field physics and quantum optics by investigating coherent polarization of ionic ensemble in the strong field condition. Furthermore, the proposed scheme also confronts the challenge of extending SC generation into UV region, facilitating more widespread applications of SC coherent light sources.

## Results

**Basic principle for SC generation.** Figure 1 schematically demonstrates the basic principle for SC generation. When nitrogen molecules are exposed in an intense mid-infrared laser field, they will be tunnel ionized. Once the electron escapes from atomic nucleus, the molecular nitrogen ion ($N_2^+$) will instantaneously feel an extremely strong electric field. Along with the switch-on of laser-ion interaction, the dynamic Stark shift driven by the intense laser field leads to continual shifts of various $N_2^+$ levels, as indicated with the shaded regions. The dynamic variation of transition energies opens up multiple five-photon resonance pathways, resulting in dipole oscillations and coherence-enhanced nonlinear polarization of ionic ensembles. The dynamic five-photon resonance occurring in multiple ionic transitions gives rise to the laser-assisted continuum emission (LACE) within the laser pulse whose wavelength depends on the transient

transition energy as well as discrete $N_2^+$ characteristic radiations due to the free-induction decay (FID) of dipole oscillations after the laser pulse. The LACE, FID signals together with fifth harmonic generation (FHG) construct the whole SC spectrum. The detailed mechanism and dynamic processes of SC generation are given in Supplementary Movie 1.

It is noteworthy that the proposed scheme is different from frequency up-conversion luminescence induced by five-photon absorption[26], in which the role of five-photon absorption is population transfer rather than inducing coherent polarization. The kind of luminescence is isotropic incoherent radiation, whereas the SC emission produced by our scheme shows a good coherence. The coherent SC radiation in the UV region is of great significance for some specific applications such as fluorescence microscopy[27], molecular structure analysis[28] and photochemical reactions monitoring[29]. As an example, we showed that the UV broadband radiation can be used as a probe to study the absorption spectrum of tunnel-ionized $CO_2$ (see the following section).

**Experimental demonstration of SC generation.** Figure 2 shows key experimental results produced by the 1580 nm, 60 fs laser pulses in the 38-mbar nitrogen gas. We first measured the UV radiation spectrum as a function of the pump laser intensity. As illustrated in Fig. 2a, the increasing pump intensity results in the distinct spectral broadening towards longer wavelengths, eventually forming a SC spectrum spanning from 300 nm to 400 nm. Unlike the SC spectrum observed in most gaseous media[30,31], the spectral broadening in current scheme is dominated by red shift with the increasing laser intensity. In addition, some narrowband radiations are superimposed on the SC spectrum, whose wavelengths remain unchanged while varying the laser intensity. These radiations are ascribed to FID signals corresponding to $B^2\Sigma_u^+(\nu') \rightarrow X^2\Sigma_g^+(\nu)$ transitions of $N_2^+$ during the field-free emission process. The continuum spectrum between two FID signals is ascribed to LACE. As illustrated in Fig. 1, both FID and LACE signals are triggered by the dynamic five-photon resonance. Note that the mechanism for LACE has not been previously identified and will be further discussed later.

Experimental results in Fig. 2a indicate that the SC generation requires a sufficiently high laser intensity. If the pump intensity is too low (e.g., $1.4 \times 10^{14}\,W\,cm^{-2}$), the UV radiation is dominated by FHG at around 320 nm. The increasing pump intensity causes the dramatic enhancement of both FID and LACE signals. At the laser intensity of $1.9 \times 10^{14}\,W\,cm^{-2}$, the 358-nm FID signal from $B^2\Sigma_u^+(\nu' = 1) \rightarrow X^2\Sigma_g^+(\nu = 0)$ transition and the fifth harmonics are bridged by the LACE to form a broad spectrum. Although the 391-nm FID signal from $B^2\Sigma_u^+(\nu' = 0) \rightarrow X^2\Sigma_g^+(\nu = 0)$ transition is also observed at this intensity, the continuum radiation is still much weaker for the wavelength above 358 nm. When the laser intensity reaches $8.7 \times 10^{14}\,W\,cm^{-2}$, the LACE between 358 nm and 391 nm is efficiently generated, forming a broad spectrum spanning ~100 nm bandwidth. At all intensities used in this experiment, the signal strength shows a sharp drop for the radiation wavelength longer than 391 nm. Besides the laser intensity, the SC radiation is also sensitive to gas pressure. Owing to the inevitable plasma defocusing in the gas chamber, the focal intensity of pump pulses will decrease with the increasing gas pressure. However, the nonlinear polarization will increase with the increasing gas density. The balance between the two effects results in the optimal gas pressure for SC generation, which is 38 mbar in our experiments.

We further compared the divergence properties of three kinds of $N_2^+$ coherent radiation (i.e., FHG, LACE, and FID). Figure 2b shows a typical angularly-resolved spectrum of SC radiation,

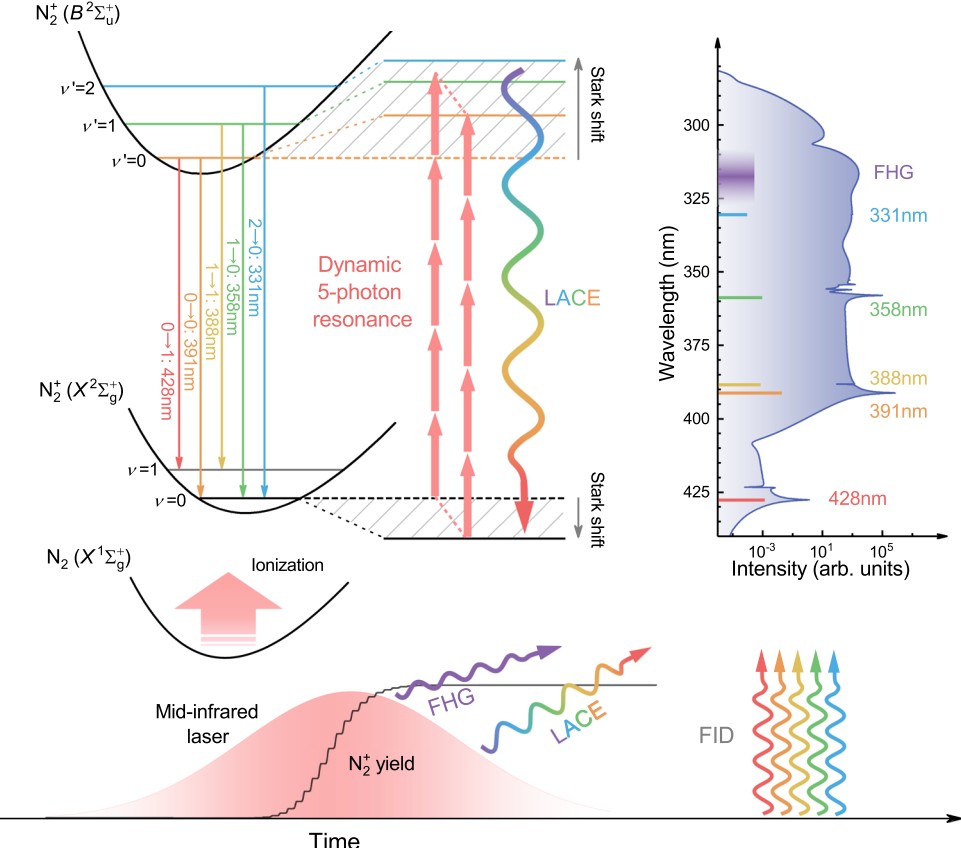

**Fig. 1 Schematic diagram of basic principle for SC generation.** Following injection of ions through ionization of neutral molecules, the dynamic Stark shift driven by an intense mid-infrared laser forces various vibrational energy levels of $B^2\Sigma_u^+$ and $X^2\Sigma_g^+$ states of $N_2^+$ to shift towards contrary directions. These instantaneously varying energy levels are indicated with shaded regions. The continual variation of transition energies with the driver laser envelope triggers dynamic five-photon resonances in multiple ionic transitions. The five-photon resonances for two special cases (i.e., without field and maximum field) are indicated with red arrows. Multi-channel five-photon resonances cause dipole oscillations and enhancement of nonlinear polarization of $N_2^+$. The coherent polarization leads to the laser-assisted continuum emission (LACE) due to continuously changing transition energies within the laser pulse as well as some narrowband $N_2^+$ characteristic radiations due to the free-induction decay (FID) of dipole oscillations after the laser pulse. These signals contribute to SC radiation together with five harmonic generation (FHG). The typical SC spectrum is shown on the right, in which the relative sizes of transition dipole moments are denoted with colored lines. (See Supplementary Movie 1 for more details).

which is measured in the polarization plane of the pump laser using an imaging spectrometer. We can clearly see that the FHG and LACE signals cannot be distinguished spectrally, and their divergence angles are close to each other. In comparison, the 391-nm FID signal is more divergent, which is in good agreement with the previous studies[15,20]. The large divergence of FID is attributed to the large phase variation of the related ionic states[15]. Therefore, the angularly-resolved measurement confirms the FID nature of these narrowband $N_2^+$ characteristic radiations. From Fig. 2b, we notice that the spectra at positive and negative angles are different. The asymmetrical spectral distribution indicates the asymmetrical spatial profile of the generated SC signal, which is mainly caused by the imperfect spatial profile of the pump laser.

**Physical mechanism of SC generation.** To understand the physical mechanism of SC generation in $N_2^+$, we constructed a theoretical model, which incorporates the photoionization and the laser-ion interaction. In our model, $N_2^+$ ions are continuously injected by tunnel ionization to participate in the laser-ion coupling. In the process of post-ionization coupling, we consider two electronic states ($X^2\Sigma_g^+$ and $B^2\Sigma_u^+$, abbreviated as $X$ and $B$, respectively) and each electronic state includes five vibrational energy levels (i.e., $v, v' = 0\sim4$). For the sake of simplicity,

we use $v'$-$v$ to mark the vibrational transition between $B$ and $X$ states. The laser parameters are chosen according to the experiment, i.e., 1580 nm, 60 fs, and the peak intensity on the order of $10^{14}$ W cm$^{-2}$. The details of theoretical simulation are given in Methods.

To get insight into the physical mechanism of SC radiation, we also presented the time-frequency properties of SC radiation by performing the wavelet transform for the induced dipole moment. The simulation result of the induced dipole moment is shown in Supplementary Fig. 1. As shown in Fig. 3a, b, the coherent radiation emerges around the peak of the laser field, in which the radiation wavelength is located in the FHG region. The dynamic Stark shift causes the transient variation of both 0-0 and 1-0 transition wavelengths. The transient transition wavelengths are obtained by calculating the cycle-averaged eigenvalues of Hamiltonian, as indicated by solid curves in Fig. 3b. When the transient transition enters the fifth harmonic region, the five-photon resonances of 0-0 and 1-0 create coherent oscillation of dipoles, which significantly change $N_2^+$ radiation properties. As we see, the evolution of coherence leads to the continuous shift of radiation towards the longer wavelength in the falling edge of the laser pulse. The kind of radiation is assigned to LACE. Its time-frequency characteristic reflects that the instantaneous Stark shift always follows the laser field. After the end of the driving pulse,

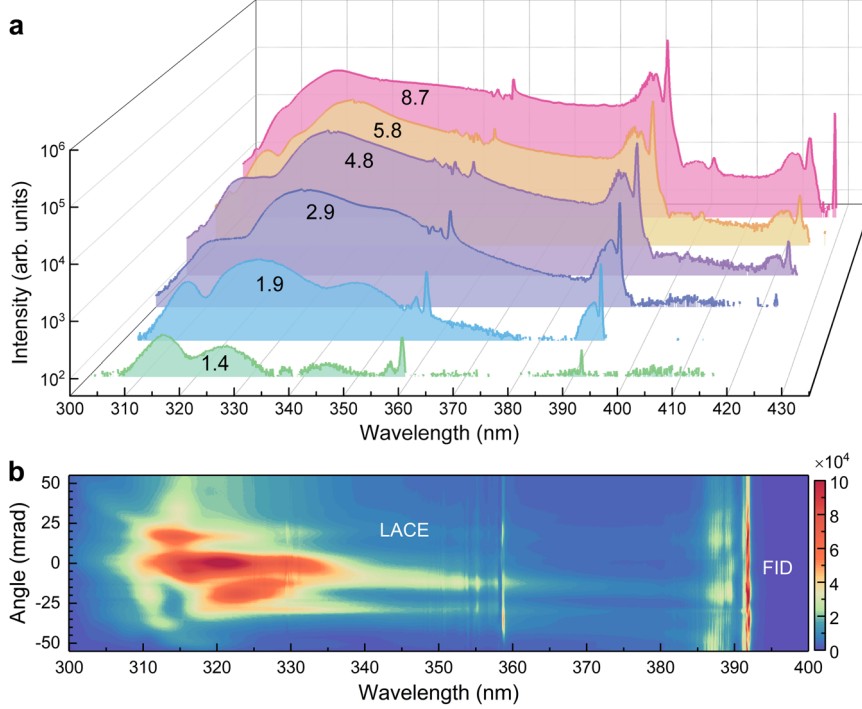

**Fig. 2 Measured SC evolution with the laser intensity and the spatial divergence characteristics of SC radiation. a** Emission spectra measured at different pump intensities. The laser intensity (unit: $10^{14}$ W cm$^{-2}$) is indicated on the corresponding spectrum. The laser intensity is estimated by measuring the size of focal spot in the case of linear propagation. **b** Typical angularly resolved spectrum of SC radiation.

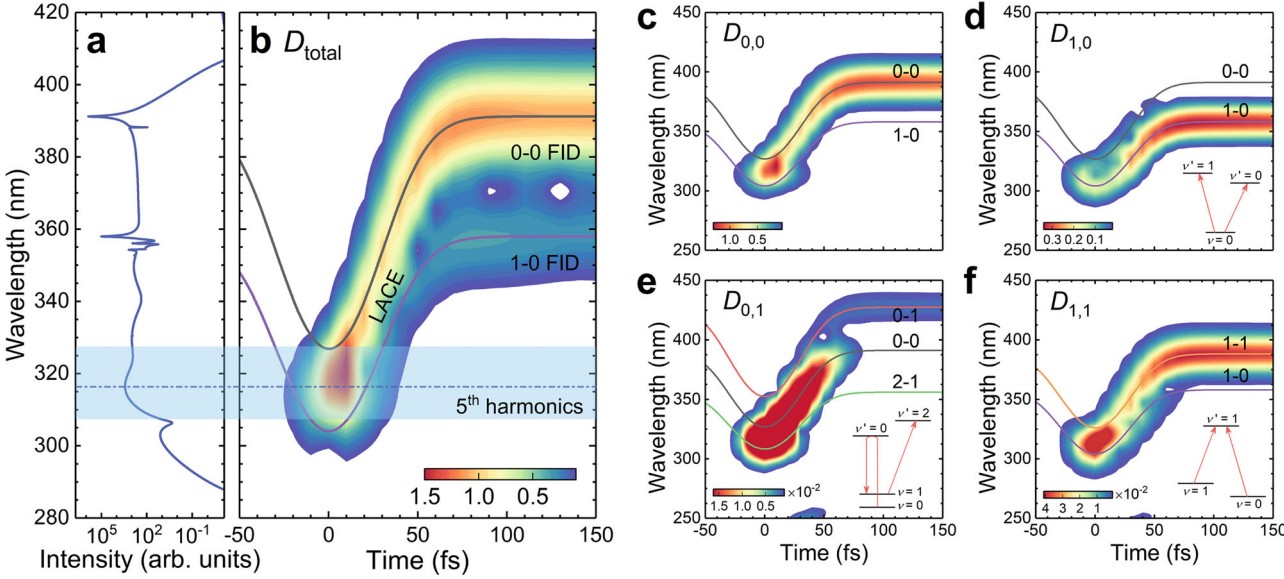

**Fig. 3 Time-frequency properties of SC radiation. a** The simulated SC spectrum with the 1580 nm, 60 fs, $2.0 \times 10^{14}$ W cm$^{-2}$ laser parameters. **b** Time-frequency structures of SC radiation obtained by performing wavelet transform for total dipole moment $D_{total}$. **c–f** Time-frequency analysis for individual dipole moment of the 0-0, 1-0, 0-1, 1-1 transition (i.e., $D_{0,0}$, $D_{1,0}$, $D_{0,1}$, $D_{1,1}$), respectively, in which transient transition wavelengths are shown by solid curves. Insets in **d–f** indicate the population transfer during the multi-channel coupling process.

all energy levels return to field-free eigenstates, while the 0-0 and 1-0 coherences are still maintained over a long time. The slow relaxation of coherence gives rise to the 358-nm and 391-nm FID signals, which are manifested as two long tails in the time-frequency diagram and narrowband radiations in the spectrum. As illustrated in Fig. 3b, the transient evolution of 0-0 and 1-0 transition wavelengths is in excellent agreement with the time-frequency structures of SC radiation, which confirms the crucial role of Stark-mediated five-photon resonance in SC generation.

The time-frequency analysis not only reveals the physical mechanism for SC generation, but also clearly illustrates the dynamic process for SC radiation. It is noteworthy that since the $N_2^+$ yield reaches the maximum around the peak of electric field, continuum radiation of ions mainly occurs in the falling edge of the driver pulse, making it easily to obtain an isolated UV ultrashort pulse after phase compensation.

As shown in Fig. 1, SC radiation arises from multi-channel multiphoton resonances. To distinguish the contribution of each

channel, we performed the time-frequency analysis for individual dipole moment. As illustrated in Fig. 3c–f, the 0-0 transition makes a dominant contribution to SC generation, while the 1-0 transition mainly gives rise to the 358-nm FID signal. However, the two emission processes are not independent of each other. As shown in the inset of Fig. 3d, the population transfer from $v = 0$ to $v' = 0$, 1 leads to the interaction of the two transition processes. Compared with the 0-0 transition, the 0-1 and 1-1 transitions are 1~2 orders of magnitude weaker due to the low population on $X(v = 1)$ state. But still all these transitions are mutually affected by each other. The 0-1 emission involves the competition of population transfer from $v = 0$ to $v' = 0$ then to $v = 1$, and that from $v = 1$ to $v' = 2$ (Fig. 3e). The population transfer paths related to the 1-1 emission are identified in Fig. 3f. Due to intercoupling and population transfer among various ionic states, we observe LACE and FID signals from other transitions in the time-frequency diagram of a given transition.

The electronic-vibrational levels participating the complex population transfer, as shown in the insets in Fig. 3d–f can be understood as V-type, Λ-type or hybrid multi-level quantum systems created in the laser-ion coupling process. The quantum optics aspect of the complex transitions is essential for SC generation. Although 0-1 and 1-0 transitions make minor contributions to total dipole moment, they play key roles in triggering five-photon resonance of the 0-0 transition. To clarify this point, we compare the dipole interaction of the laser field with two-level, Λ- and V-type three-level $N_2^+$ system, as illustrated in Fig. 4a–c. Figure 4d shows the evolution of the 0-0 transition wavelength within the laser field while different vibrational levels are involved. The corresponding emission spectra are given in Fig. 4e.

In a two-level system including $B(v' = 0)$ and $X(v = 0)$ levels, the minimum transition wavelength merely extends to 344 nm, which is far from the five-photon resonance. In this case, FHG is the main mechanism of UV radiation. In the Λ-type scheme with

the inclusion of $X(v = 1)$ level, the 0-0 transition wavelength significantly decreases as compared to the two-level case. As shown in Fig. 4b, the joint contribution of two dipole transitions forces the $B(v' = 0)$ level to shift upwards. Meanwhile, the two vibrational levels of $X$ state repel each other, leading to the down-shift of the $X(v = 0)$ level and up-shift of the $X(v = 1)$ level. The counter-shift of $B(v' = 0)$ and $X(v = 0)$ levels significantly increases the 0-0 transition energy. It is thus possible to induce SC radiation from 310 nm to 390 nm via the five-photon resonance, as evidenced by the simulated spectrum. On the contrary, in the V-type scheme, the addition of the $B(v' = 1)$ level causes a slight increase of the 0-0 transition wavelength (purple solid curve of Fig. 4d), hindering the efficient generation of the 391-nm FID signal and the spectral extension above 358 nm (Fig. 4e). As indicated in Fig. 4c, the dipole interaction among three levels makes $B(v' = 0)$ and $X(v = 0)$ levels shift towards the same direction. Thus, the addition of $B(v' = 1)$ only slightly changes the 0-0 transition energy.

The influence of vibrational excited states on the Stark shift and SC generation reflects synergistic quantum effect in the multi-level ionic system. The realistic physical process should simultaneously include all vibrational levels of $B$ and $X$ states with the considerable transition dipole moment (TDM). Thus, the generated SC radiation should be attributed to the interplay of V-type and Λ-type schemes. The transient transition wavelength and emission spectrum including all ten levels are given in Fig. 4d, e, respectively. From the result, we can clearly see that Λ-type scheme makes a greater contribution to the Stark shift and SC generation than the V-type scheme. When all vibrational levels of $B$ and $X$ states are included, the enhanced Stark shift promotes the laser-ion coupling efficiency. The produced SC spectrum in this case shows a similar spectral bandwidth with that in the Λ-type scheme. These analyses clearly demonstrate that the SC generation benefits from abundant energy levels of molecular ions.

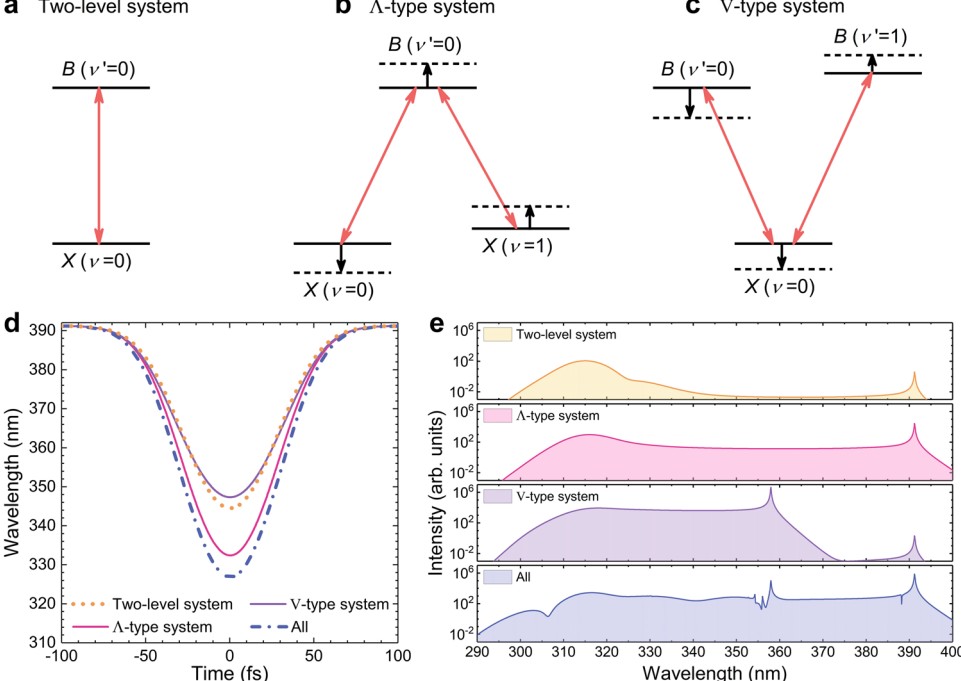

**Fig. 4 Multilevel synergistic effect in the process of dynamic Stark shift and SC generation. a–c** The dipole interaction of the laser field with two-level, Λ-type and V-type systems. Solid and dashed lines represent the energy-level positions in the two-level and three-level systems, respectively. **d, e** The transient transition wavelength of the 0-0 transition **d** and the corresponding emission spectrum **e** calculated in the two-level, Λ-type, V-type scheme and the multi-level scheme including all ten levels, respectively.

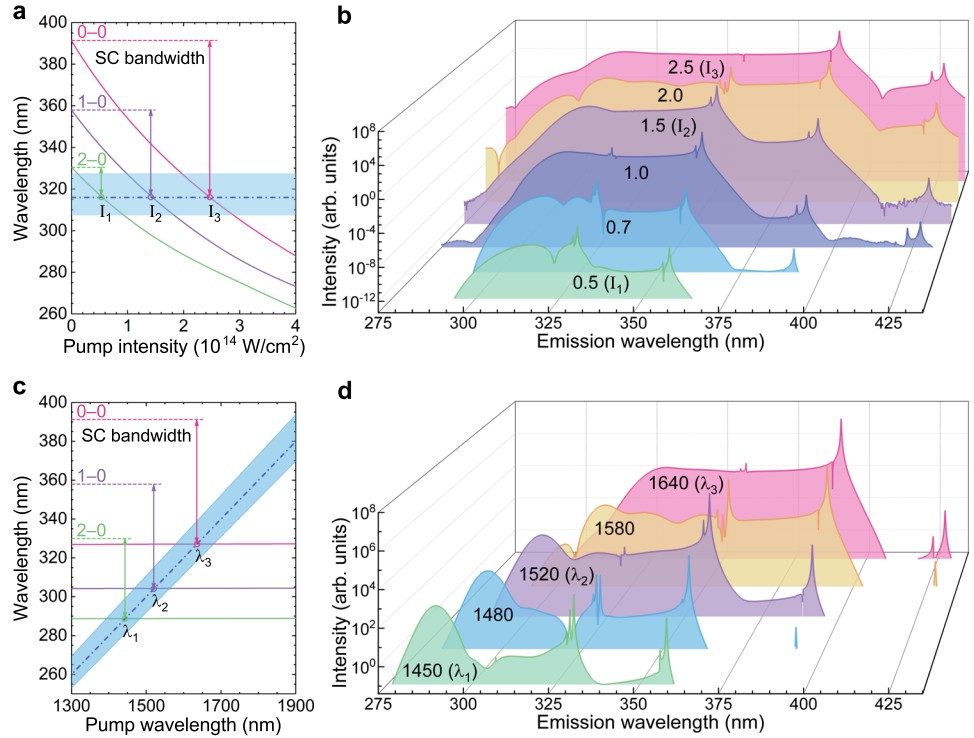

**Fig. 5 Simulated SC evolution with the pump intensity and the pump wavelength. a** The minimum transition wavelengths of three ionic transitions as a function of the 1580 nm laser intensity. The pump laser intensity (unit: $10^{14}\,\text{W cm}^{-2}$) is indicated on the corresponding spectrum. **b** The evolution of SC spectra with the 1580 nm pump laser intensity. **c** The minimum transition wavelengths of three ionic transitions as a function of the pump wavelength. At all pump wavelengths, the laser intensity and pulse duration are taken as $2.0 \times 10^{14}\,\text{W cm}^{-2}$ and 60 fs, respectively. **d** The SC spectra calculated at different pump wavelengths. The pump laser wavelength (unit: nm) is indicated on the corresponding spectrum. In both **a** and **c**, the blue-filled areas and dash-dot lines denote the spectral range and central wavelength of the fifth harmonics, respectively. The critical intensities and critical wavelengths of five-photon resonances are indicated by the cross points of solid and dash-dot curves.

**Manipulation of SC radiation**. We now focus on the manipulation of the strength and bandwidth of SC radiation. As discussed above, the Stark shift results in the transient variation of the transition wavelength between two ionic states, which strongly depends on the laser intensity. Figure 5a depicts the minimum transition wavelength as a function of the pump intensity. For a given transition, the minimum transition wavelength corresponds to maximum Stark shift (see solid curves in Fig. 3b). Here, we consider three main transitions of $N_2^+$, i.e., 0-0, 1-0, and 2-0. As we see, the transition wavelengths gradually decrease with the increasing laser intensity. When the transition enters the fifth harmonic region (blue filled area), the five-photon resonance channel is switched on accordingly. For the 2-0, 1-0, and 0-0 transition, the critical intensity triggering the resonance is $I_1 = 0.5 \times 10^{14}\,\text{W cm}^{-2}$, $I_2 = 1.5 \times 10^{14}\,\text{W cm}^{-2}$, $I_3 = 2.5 \times 10^{14}$ $\text{W cm}^{-2}$, respectively. Once five-photon resonance is triggered, strong coherence created among ionic states will induce LACE and FID. The emission spectra calculated at different intensities are given in Fig. 5b. As expected, the FID signal from the 2-0, 1-0, and 0-0 transition is efficiently produced at the corresponding critical intensity. As the laser intensity increases, all radiations are dramatically enhanced due to the strong dependence of $N_2^+$ yield and five-photon coupling on the laser intensity. Moreover, since the five-photon resonance covers a spectral range, the continuum radiation can be extended to 358 nm and 391 nm at the laser intensity of $1.0 \times 10^{14}\,\text{W cm}^{-2}$ and $2.0 \times 10^{14}\,\text{W cm}^{-2}$, respectively.

Theoretically, the bandwidth of SC radiation depends on maximum Stark shift of the triggered five-photon resonance channel. As the laser intensity increases, more resonant channels are switched on. These channels undergo different Stark shifts, as

indicated in Fig. 5a. The contribution of each channel to SC generation depends on the size of TDM and the switch-on moment of five-photon resonance. Large TDM and the multi-photon coupling at the peak of electric field are favorable for generating strong radiations in the laser-coupled ionic system. The relative size of TDM of $v'$-$v$ is indicated by colored lines in Fig. 1 and the accurate values are given in Supplementary Table 1. For three main channels discussed here, the SC radiation is dominated by the 2-0, 1-0, and 0-0 transition channels at the critical intensity of $I_1$, $I_2$, and $I_3$, respectively. The bandwidth of SC radiation is thus determined by the maximum Stark shift of the primary channel, as illustrated in Fig. 5a. As the pump intensity increases, the switch-on of new channel will cause the stepped red shift of the emission spectrum, and the red shift is more prominent than the blue shift. The asymmetrical broadening is validated by the experimental observation in Fig. 2a and the simulation result in Fig. 5b. The simulated SC evolution with the laser intensity basically follows the prediction of Fig. 5a and qualitatively agrees with the experimental result in Fig. 2a.

The pump wavelength is another key parameter to affect the laser-ion coupling and SC generation. As shown in Fig. 5c, with the aid of the dynamic Stark shift, the transition wavelengths of three channels (solid lines) significantly decrease as compared to their field-free eigenvalues (dashed lines). While scanning the pump wavelength, the five-photon resonant wavelength (dash-dot line) exhibits a linear shift, whereas the minimum transition wavelength for each channel almost remains unchanged. From Fig. 5c, we can clearly see that the five-photon resonance of the 2-0, 1-0, and 0-0 transition is triggered at the critical wavelength $\lambda_1 = 1450$ nm, $\lambda_2 = 1520$ nm and $\lambda_3 = 1640$ nm, respectively. The resonant excitation of different ionic transition channels will

affect the SC generation. Fig. 5d shows the SC spectra calculated at different pump wavelengths. As expected, the FID and LACE signals from the 2-0, 1-0, and 0-0 transition make dominant contributions to the SC radiation at the critical wavelength $\lambda_1$, $\lambda_2$, and $\lambda_3$, respectively. Since the 0-0 transition has the largest TDM, its resonant excitation will produce the strongest and broadest SC radiation. In addition, the five-photon resonance of the 0-0 transition can be triggered by both 1580 nm and 1640 nm pump lasers due to their broad spectral coverage. If the laser wavelength is too short, all five-photon resonance channels will be closed. In this case, the radiation is mainly from FHG, as observed in most gaseous media. The simulation results in Fig. 5 clearly indicate that SC generation is sensitive to the pump laser parameters. By optimizing the pump intensity and wavelength, we can obtain SC radiation with the highest strength and the broadest bandwidth.

**Application of SC radiation in the absorption spectroscopy.** The maximum energy of SC signal in our experiment is on the nJ level, and the corresponding conversion efficiency is $10^{-6}$. Although the energy is relatively low, the good coherence makes it easily to be focused to the size of <10 μm ($1/e^2$ radius). The corresponding focal intensity can reach $10^{10}$ W cm$^{-2}$ and even higher, which is sufficient to excite nonlinear effects in some condensed media. Furthermore, such broadband UV coherent radiation can also be used as a probe to study molecules or ions. Below, we will give a proof-of-principle demonstration to achieve the absorption spectrum of tunnel-ionized $CO_2$ molecules using the SC source. In the experiment, the SC radiation generated in the nitrogen gas is used as a probe, while the residual pump laser is used to ionize $CO_2$ molecules. The experimental details are given in Methods.

Figure 6 shows a typical absorption spectrum measured in the 400 mbar $CO_2$ gas. We can clearly see that the SC signal is strongly absorbed at the $\widetilde{A}^2\prod_u(\nu'_1\nu'_2\nu'_3) - \widetilde{X}^2\prod_g(\nu_1\nu_2\nu_3)$ transition wavelengths of $CO_2^+$. The main transition, denoted as $\nu'_1\nu'_2\nu'_3 - \nu_1\nu_2\nu_3$, is indicated on the corresponding absorption peak. From the absorption spectrum, we can know that after the removal of electron, most ions are populated on the ground state. In other words, the ionization from the highest occupied molecular orbital plays a dominant role under our experimental conditions. Moreover, the absorption of the 000-100 transition is much weaker than that of the 100-000 although their Franck-Condon factors are close to each other[32]. It indicates that only a small amount of ions are populated on $\widetilde{X}^2\prod_g(100)$ state. Thus, from the absorption spectrum, we can also obtain the relative

vibrational population of $\widetilde{X}^2\prod_g$ state. This experiment clearly proves that the broadband UV source can be applied as a probe in the ultrafast spectroscopy owing to its broad spectral coverage, good coherence and short pulse duration. Such broadband UV coherent light sources also have the potential to be applied in biomedical microscopy, molecular structure analysis and high-sensitivity chemical identification.

## Discussion

The current scheme for SC generation in the UV regime has several peculiar properties and advantages. First, it permits creating a SC spectrum at a relatively low pressure, which is hardly accessible in the prior investigations[33,34]. The SC radiation generated in such a condition has a better behaving time-dependent phase, making the pulse compression more easily. Second, many ionic states with proper energy gaps are exploited in our scheme, and the contributions of different ionic states to the SC generation are quite different. This evidently increases the opportunity to generate the SC with broader bandwidths and shorter wavelengths. At last, if the waveform-shaping technique is further employed to manipulate the coherence among various ionic states[35], it will be possible to generate an octave-spanning SC spectrum. It remains to be explored whether self-adaptive control of the laser waveform would help selectively enhance the LACE signal and suppress the FID signal toward a better shaped SC spectrum.

Besides promising applications, SC generation driven by ionic coherence is of great significance for the fundamental physics investigation. On the one hand, the current SC generation incorporates the tunnel ionization on the attosecond timescale, the laser-ion coupling and population transfer on the femtosecond timescale, and free induction decay of ionic coherences on the picosecond timescale. Thus, the produced SC source can serve as an intrinsic probe for interrogating multi-timescale non-equilibrium dynamics in a quantum system simultaneously including molecules and ions, which will help us gain a deeper insight into the fundamental process of intense-laser-matter interaction. On the other hand, the mechanism is not only applicable to molecular nitrogen ions but also to a range of other molecular ions. The multi-level quantum platform prepared by ultrafast lasers provides an unprecedented opportunity to investigate ionic quantum optics in the strong-field regime such as multi-electron dynamics[36–38], electron-nucleus correlation[39] and quantum coherent control[40].

In conclusion, we have demonstrated SC radiation in the UV region induced by ionic coherence in an intense laser field, wherein the dynamic five-photon resonance has been found to create the strong coherence among multiple ionic states. Such coherence gives rise to the broadband LACE signal as well as the narrowband FID signals, and their time-frequency properties record the transient variations of ionic transition energies caused by the dynamic Stark shift. Unlike the prior investigations, the SC generation in the current scheme arises from the intercoupling and synergistic effect in the multi-level quantum system, thanks to the abundant energy levels of molecular ions. In principle, the SC generation scheme can be extended to other molecular ions, opening up the possibility of SC radiation in exotic spectral regions. Besides the application in the absorption spectroscopy shown in this work, the SC source in the UV region will also inspire promising applications in biomedical photonics, high-sensitivity detection, etc. Our work also paves the way for ionic quantum optics in the strong-field regime, facilitating our understanding of the laser–matter interaction in a complex system.

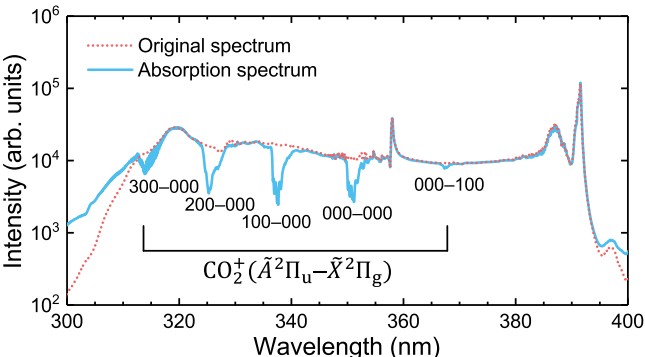

**Fig. 6 Typical $CO_2^+$ absorption spectrum measured in the 400 mbar $CO_2$ gas using the generated SC radiation as a probe.** For comparison, the original spectrum is shown with red dot line, which is captured in the case without $CO_2$.

## Methods

**Experimental details**. The experiments were performed with a high-energy optical parametric amplifier (OPA), which was pumped by a commercial Ti:sapphire laser system (Legend Elite-Duo, Coherent Inc.). The output signal beam from the OPA system was tuned to the 1580 nm wavelength. The maximum energy at the wavelength was ~1 mJ. The pump laser energy was continually adjusted with a neutral density filter. The corresponding laser intensity was estimated by measuring the focal spot under the condition of the linear propagation. Therefore, the measured intensity is higher than the realistic intensity owing to the neglected plasma defocusing effect. The pump laser was focused into a gas chamber filled with nitrogen gas. The gas pressure was optimized as 38 mbar to efficiently produce SC radiation spanning from 300 nm to 400 nm. The generated SC radiation was separated from the pump laser by a broadband dielectric mirror (BB1-E01, Thorlabs Inc.), and then was focused onto the slit of a grating spectrometer (Shamrock 303i, Andor) for spectral analysis. While measuring the angularly resolved spectrum, a cylindrical lens was used to collect the signals onto the input slit of the imaging spectrometer.

In the experiment of $CO_2^+$ absorption spectrum, the SC radiation generated in the nitrogen gas was used as a probe, while the residual pump laser was used to ionize $CO_2$ molecules. After passing through the first gas chamber filled with $N_2$ gas, they were together launched to the second chamber filled with $CO_2$. After collimation with an Aluminum-coated concave mirror, they were further focused by using another Aluminum-coated concave mirror with the focal length of 10 cm. The gas pressure in the second chamber was optimized as 400 mbar to obtain strong absorption at transition wavelengths of $CO_2^+$. The absorption and original spectra were measured in the case with and without $CO_2$ gas. The residual pump laser and the generated SC radiation passed through the fused silica window with the total thickness of 4 mm. Owing to the group velocity difference, the SC radiation in the UV regime arrives the interaction zone of the second chamber after the pump pulse for $CO_2$ ionization. The relative delay of two pulses is about 1 ps. At such a delay, the $CO_2^+$ ions have been well created by the residual pump laser, enabling the measurement of $CO_2^+$ absorption spectrum.

**Numerical calculation**. In the theoretical model, we consider the transient ionization injection and laser-ion coupling driven by the intense femtosecond laser pulses. The evolution of ionic density matrix $\rho^+(t)$ is given by[41]

$$\dot{\rho}^+ = -\frac{i}{\hbar}\left[H(t), \rho^+(t)\right] + \left[\dot{\rho}_{n\nu}^+\right]_{\text{ionize}} + \left[\dot{\rho}_{n'\nu',n\nu}^+\right]_{\text{coll}}, \quad (1)$$

where $\rho^+$ is the density matrix of $N_2^+$, $H(t)$ is the Hamiltonian, $n$ and $\nu$ denote $X^2\Sigma_g^+$ and $B^2\Sigma_u^+$ electronic states and the vibrational levels, respectively. The first and second terms in the right-hand side describe the laser-ion coupling and transient ionization, respectively. We assume that the molecular axis is aligned along the polarization direction of laser. The ionization term is expressed as $\left[\dot{\rho}_{n\nu}^+\right]_{\text{ionize}} = \Gamma_{n\nu}\rho_0$, where $\rho_0$ is the population of $N_2$ and $\Gamma_{n\nu}$ is the diagonal matrix of ionization rate, given by the MO-ADK theory[42] (see Supplementary Note 1). The coherence decay is included in the third term as $\left[\dot{\rho}_{n'\nu',n\nu}^+\right]_{\text{coll}} = -\rho_{n'\nu',n\nu}^+/T_2$, $(n'\nu' \neq n\nu)$, where the dephasing time $T_2$ is taken as 1 ps and the much slower population decay is ignored. The induced dipole moment is given by

$$D(t) = \sum_{\nu',\nu} d(B\nu', X\nu)\rho_{B\nu',X\nu}^+(t), \, (\nu',\nu = 0 \sim 4) \quad (2)$$

where $d(B\nu', X\nu)$ is TDM for a given transition $B^2\Sigma_u^+(\nu') \rightarrow X^2\Sigma_g^+(\nu)$ (see Supplementary Table 1), $\rho_{B\nu',X\nu}^+(t)$ describes the coherence between the corresponding levels. The simulation results of total induced dipole moment $D(t)$ and individual dipole moment $D_{\nu',\nu}(t)$ are given in Supplementary Note 2. The radiation spectrum is obtained by $I(\omega) = \left|\frac{1}{T}\int_{-\infty}^{+\infty}\ddot{D}(t)e^{i\omega t}dt\right|^2$ with the dipole acceleration $\ddot{D}(t)$. The time-frequency analysis is performed by the Morlet wavelet transform, which is written as

$$A(t_c, \omega) = \left|\int_{-\infty}^{+\infty} D(t)\sqrt{\omega}W\left[\omega(t - t_c)\right]dt\right|^2 \quad (3)$$

with $W(x) = \frac{1}{\sqrt{\sigma}}\exp(ix)\exp\left(-\frac{x^2}{2\sigma^2}\right)$. We choose $\sigma = 40$ fs and $t_c = 10$ fs to balance the time and frequency resolution.

## Data availability

The data that supports the plots within this paper and other findings of this study are available from the Zenodo database with https://doi.org/10.5281/zenodo.6658475 (https://zenodo.org/record/6658475#.Yq0naaJBzZv).

## Code availability

The simulation codes that support the findings of the study are available from the corresponding authors upon reasonable request.

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

## Acknowledgements

We are grateful for the financial support from the Major Research Plan of the National Natural Science Foundation of China (91850201), the National Key Research and Development Program of China (2019YFA0307703), and the National Natural Science Foundation of China (11822410, 12034013, 11874066, 12074063, 12064009, 11974426). J.Y. also acknowledges the support from Program of Shanghai Academic Research Leader (20XD1424200) and Youth Innovation Promotion Association of Chinese Academy of Sciences (2018284). H.L. acknowledges the support from Postgraduate Scientific Research Innovation Project of Hunan Province (CX20200035).

## Author contributions

J.Y. and Z.Z. discussed and conceived the idea. H.L. and J.Z. performed the simulations. J.Y. designed the experiments. F.Z., H.X., H.Z., and N.Z. performed the experiments. H.L., J.Z., Q.Z., X.W., Y.Y., J.Y. and Z.Z. analyzed the data. J.Y., H.L., H.X., G.L., L.Y., Y.C., and Z.Z. prepared the manuscript and discussed with all authors.

## Competing interests

The authors declare no competing interests.
