## [Peer Review File · Nature Communications]

Ultraviolet supercontinuum generation driven by ionic coherence in a strong laser fieldREVIEWER COMMENTS

Reviewer #1 (Remarks to the Author):

The manuscript "Ultraviolet super-continuum generation driven by ionic coherence in a strong laser field" by Hongbi Lei et al. reports the experimental observation of the broadband super-continuum spectrum in UV region from nitrogen molecular ions excited by 1580 nm light pulse and the simulated results of the theoretical model based on the time evolution of the quantum states of N₂⁺ in X and B states instantaneously prepared in the laser field using density matrix.

The experimentally observed spectra clearly show that the bandwidth of the super-continuum spectrum between 300 nm and 400 nm increases as the pump pulse intensity increases from $1.4 \times 10^{14} \text{ Wcm}^{-2}$ to $8.7 \times 10^{14} \text{ W cm}^{-2}$. The authors also present their simulation results of the time-frequency analysis of the transition dipole moment, by which they can simulate the time-evolution of the population as well as the spectra of N₂⁺ in given laser intensity. They also present that the super-continuum spectrum was initiated by the excitation from X state to B state through the fifth photon transition and followed by the strong coupling between the laser field and N₂⁺ quantum states with dynamical Stark shift in the latter half of the drive laser pulse, resulting in the broadband emission.

This work shows a potentially new and unique experimental scheme for generating broadband coherent light pulse in UV. Even though the theoretical model has been well established before, not only for researchers in quantum optics and strong laser field science but also many physicists and chemists would be interested in the presented results. I have three comments on the manuscript, and authors should consider before the publication.

1. To understand the theoretical model, I would like to know more details. In the experimental scheme, I understand the LACE is initiated by the coupling between X(0) and B(0) states by fifth harmonic generation. The authors mentioned "The dynamic variation of transition energy opens up multiple five-photon resonance pathways, resulting in dipole oscillations and coherence-enhanced nonlinear polarization of ionic ensembles", as second sentence in page 4, and in Fig. 1, the 5-photon resonance of the drive laser pulse takes place even after the Stark shift of the quantum states. I am bit confusing how the five-photon resonance contribute to the LACE process in the calculation? The theoretical model shown in Supramental, however, does seem to be one-photon coupling between the quantum states by LACE. Please explain how you take account of the five-photon resonance in the simulation and how important it is.

2. I understand the LACE is combined effect of FHG, 5-photon resonance absorption, and dynamic Stark shift induced by the drive laser pulse, but each effect should have optimized pressure. Please explain what determine the optimized nitrogen gas pressure, 38 mbar.

3. About the angular dependence of the super-continuum spectrum shown in Fig. 2b, the spectrum is shown along sagittal plane or coronal plane? I imagine that the authors measure the divergence angle with respect to the polarization direction of the drive laser pulse with certain angle. Please define or explain the definition. Also, please explain why the spectra at positive and negative angle are different to each other.

Followings are minor comments.

About the last two sentences in the first paragraph in page 2, the sentence “strong field science and quantum optics have been developing in a relatively independent way” I could not get what “relatively” means. The sentence is subjective.

In Fig. 2a, the authors should change the presentation of the super-continuum spectra so that reader understand the axis for the pump intensity is not proportional to the laser focal intensity.

Reviewer #2 (Remarks to the Author):

The manuscript “Ultraviolet supercontinuum generation driven by ionic coherence in a strong laser field” has been reviewed. The authors claimed that a novel scheme for supercontinuum generation in the UV region by combining strong-field physics and quantum optics is presented in the manuscript. The UV supercontinuum light is generated through multiphoton absorption in nitrogen ions that can survive at a relatively higher intensity. A bunch of ionic levels is involved in the supercontinuum generation by dynamic Stark-assisted multiphoton resonances following tunneling ionization. The simulation results reveal the role of multiple resonance paths, macroscopic dipole oscillations, and nonlinear polarization of the ionic ensemble. Therefore, the authors claimed that this work offers a fresh perspective for constructing exotic supercontinuum sources and opens up the territory of ionic quantum optics.

The manuscript is clearly written, and the experimental results are nicely presented at least in a technical point of view. The theoretical calculations support their experimental observation very well. However, there are two critical issues in the manuscript. First of all, it is unclear if the supercontinuum generation reported in the manuscript is completely new. For example, the frequency up-conversion

through multiphoton absorption in a multimode fiber has already been demonstrated [Phys. Rev. Appl. 14, 054063 (2020)]. The target medium was not a molecule but a solid (multimode fiber) in this work. The overall scheme of the supercontinuum generation that utilizes multiphoton absorption is very similar to the scheme presented in the current manuscript. Second, it is also unclear if the supercontinuum emission presented in the manuscript is useful as a new light source for practical applications. As explained in the manuscript, the supercontinuum emission was produced through self-phase modulation induced by ionization or the Kerr effect in photonic crystal fibers, solid material, and hollow-core fibers. These conventional techniques are highly efficient. What is the conversion efficiency in the case of LACE? Since it requires multiple steps (multiphoton absorption after tunneling), the conversion efficiency would not be so high. However, there is not information on the energy of an output pulse. It should be addressed in the manuscript.

I agree that the supercontinuum emission obtained in a molecule can be used as a probe to study molecules and ions. However, these promises should be validated separately when a specific example is demonstrated. The novelty of the scientific work and the practical usefulness as a new light source are critical points to judge the quality of the manuscript. Therefore, I cannot recommend the manuscript for publication in Nature communication.

I also have a few minor concerns listed below.

The authors mentioned, “we report a novel scheme for SC generation in the UV region by combining strong-field physics and quantum optics.” After reading this statement, I expected something related to a single photon. In fact, ‘quantum optics’ is defined on Wikipedia as “a branch of atomic, molecular, and optical physics dealing with how individual quanta of light, known as photons, interact with atoms and molecules. It includes the study of the particle-like properties of photons.” However, the supercontinuum generation presented in the manuscript is not related to the property of individual quanta of light or particle-like behavior. The supercontinuum generation is still a macroscopic response of the gas medium. Therefore, the statement made by the authors may mislead the readers.

The supercontinuum emission is produced through coherent processes such as multiphoton absorption and tunneling. Therefore, one can expect that the supercontinuum emission is also coherent, as stated in the manuscript. However, it does not directly mean that the supercontinuum emission has a useful phase structure. According to the theoretical analysis presented in the manuscript, the supercontinuum emission is linearly chirped. Therefore, the chirp of the supercontinuum emission can be compensated using chirped mirrors. However, these are shown only theoretically. It would be nice if the temporal characterization of the supercontinuum emission was included in the manuscript.

Comments from Reviewer 1 and our responses:

The manuscript “Ultraviolet super-continuum generation driven by ionic coherence in a strong laser field” by Hongbin Lei et al. reports the experimental observation of the broadband super-continuum spectrum in UV region from nitrogen molecular ions excited by 1580 nm light pulse and the simulated results of the theoretical model based on the time evolution of the quantum states of N_2^+ in X and B states instantaneously prepared in the laser field using density matrix.

The experimentally observed spectra clearly show that the bandwidth of the super-continuum spectrum between 300 nm and 400 nm increases as the pump pulse intensity increases from $1.4 \times 10^{14} \text{ W cm}^{-2}$ to $8.7 \times 10^{14} \text{ W cm}^{-2}$. The authors also present their simulation results of the time-frequency analysis of the transition dipole moment, by which they can simulate the time-evolution of the population as well as the spectra of N_2^+ in given laser intensity. They also present that the super-continuum spectrum was initiated by the excitation from X state to B state through the fifth photon transition and followed by the strong coupling between the laser field and N_2^+ quantum states with dynamical Stark shift in the latter half of the drive laser pulse, resulting in the broadband emission.

This work shows a potentially new and unique experimental scheme for generating broadband coherent light pulse in UV. Even though the theoretical model has been well established before, not only for researchers in quantum optics and strong laser field science but also many physicists and chemists would be interested in the presented results. I have three comments on the manuscript, and authors should consider before the publication.

Response: We thank the reviewer for his/her insightful suggestions and positive comments on the significance, novelty, and broad impact of our work. The reviewer’s comments and suggestions really help us improve the manuscript. The point-to-point responses are given below. We hope that our reply and the revised manuscript can successfully address the comments and questions raised by the reviewer.

1. To understand the theoretical model, I would like to know more details. In the experimental scheme, I understand the LACE is initiated by the coupling between X(0) and B(0) states by fifth harmonic generation. The authors mentioned “The dynamic variation of transition energy opens up multiple five-photon resonance pathways, resulting in dipole oscillations and coherence-enhanced nonlinear polarization of ionic ensembles”, as second sentence in page 4, and in Fig. 1, the 5-photon resonance of the drive laser pulse takes place even after the Stark shift of the quantum states. I am bit confusing how the five-photon resonance contribute to the LACE process in the calculation? The theoretical model shown in Supramental, however, does seem to be one-photon coupling

between the quantum states by LACE. Please explain how you take account of the five-photon resonance in the simulation and how important it is.

Response: Thanks a lot for this good question. First, we will describe the physical process that how the five-photon resonance contributes to the LACE. When exposed in an intense laser field, the neutral molecule will be ionized to generate nitrogen molecular ion. Meanwhile, the intense driver laser will cause transient energy level shifts of nitrogen ion due to dynamic Stark effects, and the energy shift instantaneously varies as the laser field oscillates. It means that the energy difference between excited and ground state of N_2^+ is different at different moments within the laser field. When the transition energy from ground to excited states matches with the five harmonic frequency of the driver laser, the five-photon resonance occurs. The five-photon resonance creates coherent coupling between X and B states and coherent polarization between the two states. Although the five-photon resonance could be broken in the subsequent moments owing to the change of transition energy, the formed coherent polarization can still induce the downward transition of the excited states and generate the coherent emission, of which photon energy always follows the time-dependent transient transition energy within laser field. This is so called laser-assisted continuum emission (LACE). As a result, LACE is triggered by the five-photon resonance, and “records” the transient shift of B-X transition energy induced by dynamic Stark effect. After the end of driver laser pulse, all energy levels go back to the field-free eigenstates. The residual coherent polarization will emit photons with a fixed wavelength. This is the basic process of the supercontinuum generation. To facilitate the reviewer’s understanding, we make a video (see the attached file), which has also given in the Supplementary Information.

Next, we would like to elaborate the theoretical simulation of the above process. In the simulation, the dipole approximation is used by taking the interaction potential as $V = -d \cdot E(t)$, where $E(t) = E_0(t)\cos(\omega_l t)$ is electronic field of the driver laser and d is the transition dipole moment. As shown below, **the five-photon resonance is naturally included in the interaction term without rotating wave approximation.** Taking a two-level system with the field-free eigenstates ψ_1 and ψ_2 as an example, the instantaneous eigenstates are defined as $\psi_{\pm}(t)$ and the time-dependent wave function can be written as

$$\psi(t) = c_-(t)e^{-i\int_0^t E_-(\tau)d\tau}\psi_-(t) + c_+(t)e^{-i\int_0^t E_+(\tau)d\tau}\psi_+(t). \quad (R1)$$

Here, the energies of ψ_{\pm} are given by

$$E_{\pm} = \frac{-\omega_0 \pm \sqrt{\omega_0^2 + 4|\Omega_R|^2 \cos^2(\omega_l t)}}{2}, \quad (R2)$$

where ω_0 is the field-free transition frequency and $\Omega_R = dE_0(t)/\hbar$ is the Rabi frequency. Physically, E_{\pm} are the instantaneous eigen energies associated with the instantaneous upper/lower

states $\psi_{\pm}(t)$ induced by the Stark effect. The energy difference between E_+ and E_- is the transient transition energy, i.e., $\Delta E(t) = \sqrt{\omega_0^2 + 4|\Omega_R|^2 \cos^2(\omega_l t)}$. The instantaneous eigenstates $\psi_{\pm}(t)$ is a linear superposition of ψ_1 and ψ_2 . Substituting the wave function $\psi(t)$ into the time-dependent Schrodinger equation, we obtain

$$\begin{aligned}\dot{c}_+(t) &= -\langle \psi_+ | \dot{\psi}_- \rangle e^{i\theta(t)} c_-(t) \\ \dot{c}_-(t) &= -\langle \psi_- | \dot{\psi}_+ \rangle e^{-i\theta(t)} c_+(t)\end{aligned}\quad (\text{R3})$$

with $\langle \psi_+ | \dot{\psi}_- \rangle = -\langle \psi_+ | \dot{V} | \psi_- \rangle / (E_+ - E_-) = k\omega_l \Omega_R \sin(\omega_l t)$ and $k = \omega_0 \sqrt{-E_-} / (E_+ - E_-)^{3/4}$. To get an analytical solution of transition probability, we suppose that the system is initially populated with $c_-(0) = 1$ and $c_+(0) = 0$. Considering small transition probability to the upper state, the amplitude of the lower energy state ψ_- can be considered as constants, i.e., $c_-(t) \approx 1$. Hence, we can obtain

$$c_+(t) \approx \omega_l \int_0^t k \Omega_R \sin(\omega_l \tau) e^{i\theta(\tau)} d\tau. \quad (\text{R4})$$

where $\theta(t) = \int_0^t \Delta E(\tau) d\tau$ is the accumulated phase factor. The induced coherent polarization between the instantaneous eigenstates $\psi_{\pm}(t)$ can be written as:

$$D(t) = Nd[c_+(t)c_-^*(t)e^{-i\theta(t)} + c.c.] \approx Nd[c_+(t)e^{-i\theta(t)} + c.c.]. \quad (\text{R5})$$

For a monochromatic field, the probability amplitude $c_+(t)$ can be derived as

$$c_+(t) \approx \frac{k\omega_l \Omega_R}{2} \sum_{m=-\infty}^{\infty} (-1)^m [J_m(\beta) + J_{m-1}(\beta)] \cdot \frac{1 - e^{-i[(2m-1)\omega_l - (\omega_0 + \alpha)]t}}{(2m-1)\omega_l - (\omega_0 + \alpha)}, \quad (\text{R6})$$

where $\alpha = \frac{\Omega_R^2}{\omega_0}$, $\beta = \frac{\alpha}{2\omega_l}$. It can be seen that the multiphoton resonance occurs, when $2m - 1$ order harmonic frequency is equal to the transient transition frequency induced by Stark effect, i.e., $(2m - 1)\omega_l = \omega_0 + \alpha$. Meanwhile, from Eqs. R5 and R6, we can know that the induced coherent polarization has the amplitude determined by c_+ . The additional fast oscillation phase $[e^{-i\theta(t)}]$ determines the instantaneous energy of the coherent radiation, i.e., $\frac{d\theta(t)}{dt} = \Delta E(t)$, if we assume that c_+ slowly varies with the time.

In order to examine the contribution of the five-photon resonance to laser-assisted continuum emission (LACE), we calculated the induced polarization in the two-level N_2^+ system using Eqs. R4 and R5, in which the Gaussian laser pulse is chosen. The emission spectra in the on- and off-resonance cases are plotted in Fig. R1. Clearly, when the five-photon resonance is satisfied at the driver wavelength of 1580 nm, the generated emission shows a broadband continuum spectrum owing to instantaneous change of transition energy. In contrast, when the driver wavelength is switched to 1400 nm, the five-photon resonance channel is closed. In this case, the generated fifth

harmonic wave exhibits a limited spectral bandwidth, as observed in most experiments. The comparison clearly indicates the crucial role of five-photon resonance in triggering LACE.

Fig. R1 The emission spectra in the on- (red solid line) and off-five-photon-resonance (blue dot line) cases calculated in the simplified two-level system using the pump peak intensity with 3.5×10^{14} W/cm².

Furthermore, the above analysis clearly show that the five-photon resonance significantly improves the induced coherent polarization, which emits the broadband coherent radiation. The supercontinuum generation triggered by the five-photon resonance in our work is similar to the above analysis in the two-level system, but it includes the strong-field ionization injection and the multiple coupling channels owing the participation of many vibrational energy levels. The joint contribution of multiple ionic states and the strong-field ionization injection promotes the strength and bandwidth of supercontinuum radiation.

2. *I understand the LACE is combined effect of FHG, 5-photon resonance absorption, and dynamic Stark shift induced by the drive laser pulse, but each effect should have optimized pressure. Please explain what determine the optimized nitrogen gas pressure, 38 mbar.*

Response: We agree with the reviewer's explanation. The LACE is closely related to the fifth harmonic generation, five-photon resonance and dynamic Stark shift. All these effects depend on the driver laser intensity. As shown by experimental observations and theoretical simulations, the LACE cannot be efficiently produced if the laser intensity is insufficient to trigger the five-photon resonance. Our experiment is performed in a static gas chamber filled with nitrogen gas. Owing to the inevitable plasma defocusing, the focused intensity of driver pulses will decrease with the increasing gas pressure. However, the macroscopic nonlinear polarization will increase with the increasing gas density. The balance between the two effects will result in the optimal gas pressure at which LACE is the strongest.

We have added a brief explanation on this optimized gas pressure in our revised manuscript (See paragraph 1 of page 7).

3. About the angular dependence of the super-continuum spectrum shown in Fig. 2b, the spectrum is shown along sagittal plane or coronal plane? I imagine that the authors measure the divergence angle with respect to the polarization direction of the drive laser pulse with certain angle. Please define or explain the definition. Also, please explain why the spectra at positive and negative angle are different to each other.

Response: Thanks a lot for the helpful comment. The angularly-resolved spectrum in Fig. 2(b) is measured in the plane perpendicular to the propagation direction of the driver laser. In this work, the UV supercontinuum radiation is generated near focus as the secondary radiation of the driver laser. Thus, its divergence is closely related to the focusing geometry. As shown in Fig. R2, the divergent supercontinuum radiation after focus is collimated by a concave mirror with a focal length $f=50$ mm. The divergence angle θ is calculated by $\theta = \text{atan}(\frac{R-R_0}{f})$. To enhance the collection efficiency, a cylindrical lens was used to collect the signals onto the input slit of the imaging spectrometer. As shown in Fig. R2, the beam focused by a cylindrical lens is parallel to the slit. Thus, the angularly resolved spectrum recording in the imaging spectrometer shows the spectral structure for different θ angles.

Fig. R2 The experimental setup for measuring the angularly-resolved spectrum.

In addition, the asymmetrical spectral distribution at positive and negative angles originates from the asymmetrical spatial profile of the generated supercontinuum signal. Actually, the driver laser beam from optical parameter amplifier is not a Gaussian beam. The spatial asymmetry of the driver laser results in the asymmetrical spatial profile of the supercontinuum radiation.

Following the reviewer's advice, we have added the description of divergence angle measurement and the explanation on the asymmetry of angularly-resolved spectrum in our revised manuscript (see paragraph 2 of page 7).

Followings are minor comments.

About the last two sentences in the first paragraph in page 2, the sentence "strong field science and quantum optics have been developing in a relatively independent way" I could not get what "relatively" means. The sentence is subjective.

Response: Thanks a lot for the reviewer's suggestion. Following the reviewer's suggestion, the sentence has been changed as "*Over the past decades, interdisciplinary studies of strong field physics and quantum optics have been rarely explored owing to fundamental differences in their focused subjects and the used laser parameters*" (see paragraph 1 of page 3).

In Fig. 2a, the authors should change the presentation of the super-continuum spectra so that reader understand the axis for the pump intensity is not proportional to the laser focal intensity.

Response: Thanks a lot for the reviewer's suggestion. We have removed the axis for pump intensity in the supercontinuum spectra to avoid any misunderstanding in the revised manuscript. Figures 5(b) and (d) have also been modified accordingly.

Comments from Reviewer 2 and our responses:

The manuscript "Ultraviolet supercontinuum generation driven by ionic coherence in a strong laser field" has been reviewed. The authors claimed that a novel scheme for supercontinuum generation in the UV region by combining strong-field physics and quantum optics is presented in the manuscript. The UV supercontinuum light is generated through multiphoton absorption in nitrogen ions that can survive at a relatively higher intensity. A bunch of ionic levels is involved in the supercontinuum generation by dynamic Stark-assisted multiphoton resonances following tunneling ionization. The simulation results reveal the role of multiple resonance paths, macroscopic dipole oscillations, and nonlinear polarization of the ionic ensemble. Therefore, the authors claimed that this work offers a fresh perspective for constructing exotic supercontinuum sources and opens up the territory of ionic quantum optics.

The manuscript is clearly written, and the experimental results are nicely presented at least in a technical point of view. The theoretical calculations support their experimental observation very well. However, there are two critical issues in the manuscript. First of all, it is unclear if the supercontinuum generation reported in the manuscript is completely new. For example, the

frequency up-conversion through multiphoton absorption in a multimode fiber has already been demonstrated [Phys. Rev. Appl. 14, 054063 (2020)]. The target medium was not a molecule but a solid (multimode fiber) in this work. The overall scheme of the supercontinuum generation that utilizes multiphoton absorption is very similar to the scheme presented in the current manuscript.

Response: We appreciated the reviewer's recognition on writing, experimental demonstration and theoretical simulation. The main criticisms of the reviewer are the novelty of supercontinuum generation mechanism and its application. We will give some detailed explanations for the two points. We hope that our explanation can fully address the reviewer's concerns so that he/she can re-consider the decision.

The reviewer mentioned that the supercontinuum generation mechanism in our work is similar to multimode fiber luminescence [Phys. Rev. Appl. 14, 054063 (2020)], and he/she thus argues the novelty of our method. Actually, our proposed mechanism for generating laser-assisted continuum emission (LACE) is fundamentally different from that of the frequency up-conversion luminescence in a multimode fiber, although the similar strategy of five-photon resonance is adopted.

In the work of multimode fiber luminescence, the supercontinuum light is **luminescence emitted by the intrinsic and induced defects in the multimode fiber**. As shown in Fig. R3(a), the electrons in the low energy band S_0 can be excited to the high energy band S_1 with the aid of the five-photon resonance, which later relaxes to an intermediate energy band T_1 via the electron-photon scattering. **The up-conversion luminescence takes place via the spontaneous decay from the excited energy band T_1 to ground energy band S_0 , as shown in Fig. R3(a). Therefore, the five-photon absorption involved in this scheme just plays a role of population transfer, rather than directly inducing supercontinuum generation. The emission wavelength and bandwidth depend on the energy difference between two energy bands. Moreover, this kind of luminescence is incoherent radiation, which cannot be used to generate the ultrafast and strong-field pulses**

However, in our case, the UV supercontinuum radiation originates from ionic nonlinear polarization triggered by five photon resonance. In this process, both population transfer and ionic polarization is coherent, and thus the supercontinuum radiation is coherent, which yields the fundamental difference and novel significance compared with the up-conversion luminescence. Especially, the coherent supercontinuum is mainly emitted under the driving of the laser pulse. The generation mechanism is schematically shown in Fig. R3(b). When exposed in an intense laser field (pink Gaussian pulse), the neutral molecule will be ionized to generate nitrogen molecular ion. Meanwhile, the intense driver laser will cause dynamic Stark effect. The Stark effect results in the down-shift of the lower state $X^2\Sigma_g^+$ and up-shift of the upper state $B^2\Sigma_u^+$.

Thus, the transition energy instantaneously varies as the laser field oscillates. It means that the energy difference between excited and ground state of N_2^+ is different at different moments within the laser field. When the transition energy from ground to excited states matches with the five harmonic frequency of the driver laser, the five-photon resonance occurs. The five-photon resonance creates coherent coupling between X and B states and coherent polarization between the two states. Afterwards, the coherent polarization emits photons, whose frequency is equal to the transient transition energy. It means that the emission wavelength continually varies within the laser field. As a result, the supercontinuum radiation originates from the joint contribution of strong field ionization, five-photon excitation and dynamic Stark shift. **Therefore, in our scheme, the key role of five-photon resonance is to induce coherent polarization.** Furthermore, the emission wavelength and bandwidth of supercontinuum radiation depends on the transient energy difference between excited and ground states. **The generated emission shows a small divergence,** as shown in Fig. 2(b) of main text, which is significantly different from the luminescence.

Fig. R3 (a) Schematic diagram of up-conversion luminescence induced by five-photon resonant absorption. (b) Schematic diagram of laser-induced continuum emission (LACE) induced by Stark-shift-assisted five-photon resonance.

To illustrate the concept above and eliminate possible confusions, we also make a video in the multi-level system in the attached file. The video has also been added in the Supplemental Information as a separated file.

Based on the above analysis, we think that the supercontinuum generation mechanism in our work is fundamentally different from the work mentioned by the reviewer. Furthermore, **we also would like to emphasize the novelty of our medium.** Here, we choose molecular nitrogen ion produced by tunnel ionization as the target medium for the following reasons. Firstly, compared with neutral atoms/molecules, the ions can withstand a higher laser intensity, allowing for larger energy shifts

as well as broader, stronger supercontinuum radiation. Secondly, the molecular ion usually possesses abundant energy vibrational and rotational levels, which can be coupled together in an intense laser field. The joint contributions of multiple ionic states evidently increase the opportunity to generate the supercontinuum with broader bandwidths and shorter wavelengths. **Therefore, we believe that our work is novel in terms of supercontinuum generation scheme and the selected quantum system.**

To highlight the novelty and significance of our work, we have added the comparison of our method with up-conversion luminescence in our revised manuscript (see page 4).

Second, it is also unclear if the supercontinuum emission presented in the manuscript is useful as a new light source for practical applications. As explained in the manuscript, the supercontinuum emission was produced through self-phase modulation induced by ionization or the Kerr effect in photonic crystal fibers, solid material, and hollow-core fibers. These conventional techniques are highly efficient. What is the conversion efficiency in the case of LACE? Since it requires multiple steps (multiphoton absorption after tunneling), the conversion efficiency would not be so high. However, there is not information on the energy of an output pulse. It should be addressed in the manuscript.

Response: Thanks a lot for the reviewer's insightful suggestions and comments. As compared to the supercontinuum emission produced in photonic crystal fibers, solid material and hollow fibers, the supercontinuum emission obtained by our scheme is relatively weaker, owing to smaller nonlinear coefficient, lower gas density and shorter interaction length. **The maximum energy of the supercontinuum signal is measured to 2 nJ in our experiment. The corresponding conversion efficiency is estimated as 2×10^{-6} .**

Although the supercontinuum radiation is not so strong, it still has the following advantages. First, the generation scheme can be extended to vacuum ultraviolet (VUV) region, if the suitable molecular ions with larger energy gap (e.g., CO^+) are selected. The VUV supercontinuum is of great significance for exploring ultrafast dynamic of atoms, molecules and condensed matter, but such source has not been easily accessible. Our scheme offers a feasible solution. More importantly, the LACE in the VUV and EUV regimes can provide a new method for isolated attosecond pulse generation. Second, the spectral bandwidth and phase of SC radiation is easily controlled by the pulse shaping of driver laser. Third, due to the good spatial coherence, the supercontinuum signal can be focused to $5 \sim 6 \mu\text{m}$ ($1/e^2$ radius) using a lens with a focal length of 36 mm, as shown in Fig. R4. Assuming the pulse duration is 50 fs, the corresponding **focal intensity can reach $4 \times 10^{10} \text{ W/cm}^2$** . The laser intensity is high enough to excite nonlinear effects in some condensed media.

Following the reviewer's suggestion, we have added the information of energy, conversion efficiency of the supercontinuum radiation and the related discussion in the revised manuscript (see pages 15-16).

Fig. R4 The measured focal spot of the supercontinuum radiation by using a lens with the focal length of 36 mm.

I agree that the supercontinuum emission obtained in a molecule can be used as a probe to study molecules and ions. However, these promises should be validated separately when a specific example is demonstrated. The novelty of the scientific work and the practical usefulness as a new light source are critical points to judge the quality of the manuscript. Therefore, I cannot recommend the manuscript for publication in Nature communication.

Response: As the reviewer pointed out, the supercontinuum radiation can be used as a probe to study molecules or ions. Instead of the multimode fiber luminescence, the LACE radiation is more akin to the extreme ultraviolet (XUV) continuum [Nat. Photonics, 4, 875-879, (2010); Opt. Express, 25, 27506, (2017); Nat. Commun., 8, 186, (2017)] generated through high-order harmonic generation (HHG) in terms of coherence and conversion efficiency (or pulse energy). With higher photon energy, the XUV continuum is successfully applied to probe excitation and ionization dynamics in atomic/molecular systems [Sci. Rep., 3, 115, (2013); Nat. Commun., 11, 5042, (2020)] with pump-probe based transient absorption technique. The LACE radiation is by contrast suitable to probe vibrational and rotational dynamics of molecules by developing similar transient absorption apparatus. Here, we show the transient absorption spectroscopy measurement of CO_2^+ with our LACE source. We demonstrate the absorption spectra measurement and vibrational state identifying, but leave the dynamics undiscussed, since it is out of the scope of the manuscript. The experimental setup and result are shown in Fig. R5 and Fig. R6, respectively.

In the experiment, the SC radiation generated in the nitrogen gas is used as a probe, while the residual pump laser is used to ionize CO_2 molecules. After passing through the first gas chamber filled with N_2 gas, they are together launched to the second chamber filled with CO_2 . After collimation with an Aluminum-coated concave mirror (CM1), they are further focused by using

another concave mirror with the focal length of 10 cm (CM2). The gas pressure in the second chamber is optimized as 400 mbar to obtain strong absorption at transition wavelengths of CO_2^+ . The absorption and original spectra are measured in the case with and without CO_2 gas. The residual pump laser and the generated SC radiation pass through the fused silica window with the total thickness of 4 mm. Owing to the group velocity difference, the SC radiation in the UV regime arrives the interaction zone of the second chamber after the driver pulse for CO_2 ionization. The relative delay is estimated as 1.2 ps. At such a delay, the CO_2^+ ions have been well created by the residual pump laser, enabling the measurement of CO_2^+ absorption spectrum.

Fig. R5 Experimental setup of the CO_2^+ absorption spectroscopy.

Figure R6 shows a typical absorption spectrum measured in the 400 mbar CO_2 gas. We can clearly see that the SC signal is strongly absorbed at the $\tilde{A}^2\Pi_u(v'_1v'_2v'_3) - \tilde{X}^2\Pi_g(v_1v_2v_3)$ transition wavelengths of CO_2^+ . The main transition, denoted as $v'_1v'_2v'_3 - v_1v_2v_3$, is indicated on the corresponding absorption peak. From the absorption spectrum, we can know that after the removal of electron, most ions are populated on the ground state. In other words, the ionization from the highest occupied molecular orbital plays a dominant role under our experimental conditions. Moreover, the absorption of the 000 – 100 transition is much weaker than that of the 100 – 000 although their Franck-Condon factors are close to each other. It indicates that only a small amount of ions are populated on $\tilde{X}^2\Pi_g(100)$. Thus, from the absorption spectrum, we can also obtain the relative vibrational population of $\tilde{X}^2\Pi_g$ state.

Fig. R6 Typical absorption spectrum of CO_2^+ measured in 400 mbar CO_2 gas.

The above experiment clearly proves that the broadband UV source can be applied as a probe in the ultrafast spectroscopy owing to its broad spectral coverage, good coherence and short pulse duration. Such broadband UV coherent light sources are also expected to apply in biomedical microscopy, molecular structure analysis and high-sensitivity chemical identification in the future.

We have demonstrated the application of our supercontinuum source in the absorption spectroscopy of CO_2^+ in our revised manuscript (See Figure 6 and the relevant discussion on pages 16-17).

I also have a few minor concerns listed below.

The authors mentioned, “we report a novel scheme for SC generation in the UV region by combining strong-field physics and quantum optics.” After reading this statement, I expected something related to a single photon. In fact, ‘quantum optics’ is defined on Wikipedia as “a branch of atomic, molecular, and optical physics dealing with how individual quanta of light, known as photons, interact with atoms and molecules. It includes the study of the particle-like properties of photons.” However, the supercontinuum generation presented in the manuscript is not related to the property of individual quanta of light or particle-like behavior. The supercontinuum generation is still a macroscopic response of the gas medium. Therefore, the statement made by the authors may mislead the readers.

Response: Thanks for the comments. We would like to emphasize that the concept of quantum optics discussed in our manuscript is more concerned with a quantized molecular system interacting with a classical light field, which is dealt with the semi-classical theory. The semi-classical approach

using density matrix formalism is still within the scope of quantum optics, as can be found in the famous textbook of “*Quantum Optics*” (M. Scully and M. Zubairy, Cambridge University, 1997), which can be successfully used to explain the well-known phenomena including electromagnetically induced transparency (see page 225), lasing without inversion (see page 230).

Of course, we fully agree with the reviewer that in traditional quantum optics, most studies, especially relating to the quantum information or computation, require the quantization of the light field, in which the laser field is extremely weak. However, in our case, the molecular ionic system is driven via a strong laser field and the coherence of the system prepared by a multi-photon process can also lead to the similar observations of many quantum optical phenomena, such as electromagnetically induced transparency, lasing without inversion, slow light, etc. In this work, we report a novel scheme for supercontinuum generation in the UV region by combining strong-field physics and quantum optics in a broad sense.

The supercontinuum emission is produced through coherent processes such as multiphoton absorption and tunneling. Therefore, one can expect that the supercontinuum emission is also coherent, as stated in the manuscript. However, it does not directly mean that the supercontinuum emission has a useful phase structure. According to the theoretical analysis presented in the manuscript, the supercontinuum emission is linearly chirped. Therefore, the chirp of the supercontinuum emission can be compensated using chirped mirrors. However, these are shown only theoretically. It would be nice if the temporal characterization of the supercontinuum emission was included in the manuscript.

Response: The supercontinuum emission produced by our method shows a good spatial coherence. As shown in Fig. 2(b), its divergence angle is smaller than the driver laser, indicating that it has a good directionality or spatial coherence, as the reviewer predicted. However, the time-dependent phase structure is determined by its generation mechanism. As explained above, the wavelength of supercontinuum radiation in our scheme always follows the transient transition energy. The transient energy shift is the result of dynamic Stark effect, as evidenced by Fig. 3(b). Thus, the supercontinuum emission is nearly linearly chirped, which is easily compensated using chirped mirrors. We fully agree with the reviewer that the experimental measurement of the temporal characteristic will be helpful for the pulse compression of the supercontinuum source. Actually, we attempt to perform the measurement using cross correlation technique. However, to obtain measurable sum-frequency signals of the supercontinuum radiation and the reference pulse at 800 nm, BBO crystal must be sufficiently thick. In this case, the phase-matched bandwidth cannot support such broad spectral range. We expected that the problem can be solved using a high-sensitivity detector (e.g., EMCCD, ICCD) and a thin nonlinear crystal with a broad phase-matched

bandwidth in the future. Although the temporal characteristic of the supercontinuum emission cannot be shown experimentally in the current stage, it will not affect the novelty and integrality.

In the revised manuscript, we have shown the fundamentally difference of the proposed supercontinuum scheme with the up-conversion luminescence, emphasizing the novelty of our method. Furthermore, we also show the application of such a supercontinuum source in the absorption spectroscopy of CO_2^+ . We hope that the reviewer is satisfied with the revised manuscript.

The list of changes:

Note: Following suggestions and comments of two reviewers, we have revised the manuscript. The changes made on the manuscript have also been highlighted in blue in the revised manuscript.

1. Following the reviewer 1's suggestion, we have modified the sentence "strong field physics and quantum optics have been developing in a relatively independent way" (see paragraph 1 of page 3).
2. Following the reviewer 1's suggestion, we have added a brief explanation on this optimized gas pressure in our revised manuscript (paragraph 1 of page 7).
3. Following the reviewer 1's advice, we have added the description of divergence angle measurement and the explanation on the asymmetry of angularly-resolved spectrum in our revised manuscript (see paragraph 2 of page 7).
4. Following the reviewer 1's advice, we have removed the axis for pump intensity in the supercontinuum spectra to avoid any misunderstanding in the revised manuscript. Figure 5(b) and (d) has also been modified accordingly. The relevant descriptions are given in the corresponding caption.
5. To highlight the novelty and significance of our work, we have added the comparison of our method and up-conversion luminescence in our revised manuscript (see page 4) and the reference the reviewer mentioned has been added (Ref. 26).
6. Following the reviewer 2's suggestion, we have added the description on the energy, conversion efficiency of supercontinuum radiation in the revised manuscript (see paragraph 2 of page 15 and paragraph 1 of page 16).
7. Following the reviewer 2's suggestion, we have demonstrated the application of our supercontinuum source in the absorption spectroscopy of CO_2^+ in our revised manuscript (see Figure 6 and the relevant discussion, pages 16-17). We also modified the abstract and conclusion accordingly, and added Ref. 32.
8. We have added the experimental details on the absorption spectroscopy of CO_2^+ in Methods.
9. We have simplified the abstract due to the word limit.
10. We have added two authors due to their great contribution in performing new experiments. The contributions of authors have also been modified accordingly.
11. We have modified the theoretical description in the supplemental information and added a video in the supplemental information.

REVIEWERS' COMMENTS

Reviewer #1 (Remarks to the Author):

The authors well responded to my comments and the revised manuscript "Ultraviolet supercontinuum generation driven by ionic coherence in a strong laser field," by Hongbin Lei et al becomes much better now. Even without the demonstration of CO₂⁺ absorption spectrum, the revised manuscript is worth publishing, but the demonstration is very impressive to me. Here I recommend the revised manuscript to be published in Nature Communications.

I have very minor comment regarding the additional description in page 16.

I would suggest to revise the expression in line 339-341 as follow:

The corresponding focal intensity can reach 10^{10} W cm⁻² and even higher, which is sufficient to excite nonlinear effect in some condensed media.

Reviewer #2 (Remarks to the Author):

The revised manuscript has been significantly improved during the review process. The authors have answered all my concerns clearly. In particular, one of the critical issues related to novelty has been solved now. I believe that the new scheme for supercontinuum generation presented in the manuscript will be of interest to researchers who study molecular dynamics in a strong laser field. Therefore, I do not object to the publication of the manuscript, considering the novelty of the work. However, it is a pity that the conversion efficiency is only 10^{-6} . The application of the new scheme for supercontinuum generation is very limited. Thus I cannot highly recommend the manuscript for publication in Nature Communication.

Comments from Reviewer 1 and our responses:

The authors well responded to my comments and the revised manuscript "Ultraviolet supercontinuum generation driven by ionic coherence in a strong laser field," by Hongbin Lei et al becomes much better now. Even without the demonstration of CO_2^+ absorption spectrum, the revised manuscript is worth publishing, but the demonstration is very impressive to me. Here I recommend the revised manuscript to be published in Nature Communications.

Response: We are glad that our responses satisfy the reviewer. We are also very grateful to the reviewer for his/her great efforts in reviewing our manuscript and recommendation.

I have very minor comment regarding the additional description in page 16.

I would suggest to revise the expression in line 339-341 as follow:

The corresponding focal intensity can reach $10^{10} \text{ W cm}^{-2}$ and even higher, which is sufficient to excite nonlinear effect in some condensed media.

Response: We thank the reviewer for his/her careful review for our manuscript. Based on his/her suggestions, we have modified the relevant description in our revised manuscript.

Comments from Reviewer 2 and our responses:

The revised manuscript has been significantly improved during the review process. The authors have answered all my concerns clearly. In particular, one of the critical issues related to novelty has been solved now. I believe that the new scheme for supercontinuum generation presented in the manuscript will be of interest to researchers who study molecular dynamics in a strong laser field. Therefore, I do not object to the publication of the manuscript, considering the novelty of the work. However, it is a pity that the conversion efficiency is only 10^{-6} . The application of the new scheme for supercontinuum generation is very limited. Thus I cannot highly recommend the manuscript for publication in Nature Communication.

Response: We appreciated the reviewer's recognition on the novelty of our work after reviewing our manuscript again. We agree with the reviewer that the low conversion efficiency of the supercontinuum generation in the proposed scheme will limit its potential for more broad applications. Indeed, the quantum efficiency of generating coherence sources, such as high harmonics and supercontinuum, is intrinsically low for gas medium when comparing with solids. However, we would like to stress that besides being applied to molecular dynamics as a probe, such supercontinuum coherent light source is already capable of exciting nonlinear effects in some condensed media owing to its high focal intensity up to $10^{10} \text{ W cm}^{-2}$. In addition, we also believe that the conversion efficiency of the supercontinuum radiation can be further enhanced by the choice of quantum system and the optimization of laser-ion coherent couplings by using the waveform-

shaping technique, which will be investigated in the future work.